# Master Regulators of Causal Networks in Intestinal- and Diffuse-Type Gastric Cancer and the Relation to the RNA Virus Infection Pathway

**DOI:** 10.3390/ijms25168821

**Published:** 2024-08-13

**Authors:** Shihori Tanabe, Sabina Quader, Horacio Cabral, Edward J. Perkins, Hiroshi Yokozaki, Hiroki Sasaki

**Affiliations:** 1Division of Risk Assessment, Center for Biological Safety and Research, National Institute of Health Sciences, Kawasaki 210-9501, Japan; 2Innovation Centre of NanoMedicine (iCONM), Kawasaki Institute of Industrial Promotion, Kawasaki 210-0821, Japan; 3Department of Bioengineering, Graduate School of Engineering, The University of Tokyo, Tokyo 113-0033, Japan; 4US Army Engineer Research and Development Center, Vicksburg, MS 39180, USA; 5Department of Pathology, Kobe University Graduate School of Medicine, Kobe 650-0017, Japan; 6Department of Pharmacology and Therapeutics, National Cancer Center Research Institute, Tokyo 104-0045, Japan

**Keywords:** causal network, epithelial–mesenchymal transition (EMT), gastric cancer (GC), molecular network, microRNA (miRNA)

## Abstract

Causal networks are important for understanding disease signaling alterations. To reveal the network pathways affected in the epithelial–mesenchymal transition (EMT) and cancer stem cells (CSCs), which are related to the poor prognosis of cancer, the molecular networks and gene expression in diffuse- and intestinal-type gastric cancer (GC) were analyzed. The network pathways in GC were analyzed using Ingenuity Pathway Analysis (IPA). The analysis of the probe sets in which the gene expression had significant differences between diffuse- and intestinal-type GC in RNA sequencing of the publicly available data identified 1099 causal networks in diffuse- and intestinal-type GC. Master regulators of the causal networks included lenvatinib, pyrotinib, histone deacetylase 1 (HDAC1), mir-196, and erb-b2 receptor tyrosine kinase 2 (ERBB2). The analysis of the HDAC1-interacting network identified the involvement of EMT regulation via the growth factors pathway, the coronavirus pathogenesis pathway, and vorinostat. The network had RNA–RNA interactions with microRNAs such as mir-10, mir-15, mir-17, mir-19, mir-21, mir-223, mir-25, mir-27, mir-29, and mir-34. The molecular networks revealed in the study may lead to identifying drug targets for GC.

## 1. Introduction

Various diseases have different molecular network dynamics. A number of signaling pathways and molecular networks have several activation statuses in diffuse- and intestinal-type gastric cancer (GC) [1,2,3]. For instance, the epithelial–mesenchymal transition (EMT) regulation pathway alters in diffuse- and intestinal-type GC [1,2]. The main features of EMT include the acquisition of anti-cancer drug resistance, recurrence, and metastasis of cancer. EMT and cancer stem cells (CSCs) share some properties in terms of anti-cancer drug resistance. The precise mechanism of how CSCs arise and the condition of the EMT’s appearance are unknown.

Some of the cells with drug resistance exhibit EMT features. The main population causing the drug resistance comprises CSCs that have a high expression of transporters such as ATP binding cassette (ABC) transporters [4,5,6]. EMT, itself, is related to the development process induced by TGF-beta [7] and shares some properties with CSCs in terms of cancer malignancy, such as metastasis, recurrence, and drug resistance [8].

EMT and resistance toward therapeutics characterize metastatic GC [9]. The molecular pathways related to EMT in metastatic and resistant GC have been identified. However, the causal networks in GC with poor prognosis are poorly understood [10,11,12]. The diffuse-type GC has a much poorer prognosis compared to the intestinal-type GC. A previous study revealed that a significant difference exists in the spread patterns of intestinal- and diffuse-type GC [13]. The identification of the regulators of molecular networks in diffuse- and intestinal-type GC is crucial for understanding drug resistance and finding therapeutic targets for diffuse-type GC [14].

In this study, we investigated the master regulators of the causal networks in diffuse- and intestinal-type GC and found some interactions of molecular networks in diffuse- and intestinal-type GC. The study elucidated an interesting relationship between diffuse- and intestinal-type GC molecular networks. These findings may be useful for targeting the new therapeutics in metastatic GC characterized by EMT. Considering that EMT and drug resistance share some characteristics, we performed a close investigation of molecular networks in diffuse- and intestinal-type GC.

## 2. Results

### 2.1. Master Regulators of 1099 Causal Networks in Diffuse- and Intestinal-Type Gastric Cancer

The upstream analysis of the gene expression data in diffuse- and intestinal-type GC identified 1099 causal networks, which include the master regulators shown in Figure 1. The analysis identified lenvatinib, pyrotinib, histone deacetylase 1 (HDAC1), microRNA (mir)-196, and erb-b2 receptor tyrosine kinase 2 (ERBB2) as master regulators of causal networks in diffuse- and intestinal-type GC (Figure 1). Lenvatinib and pyrotinib were the chemical drugs of molecule type, HDAC1 was the transcription regulator of molecule type, mir-196 was the microRNA of molecule type, and ERBB2 was the kinase of molecule type (Figure 1a–d). The causal network of lenvatinib included 718 molecules, and the predicted activation z-score was 7.69 in diffuse-type GC (Figure 1a). The causal network of pyrotinib included 188 molecules, and the predicted activation z-score was 6.86 in diffuse-type GC, whereas in intestinal-type GC, it was −3.50 (Figure 1a). The causal network of HDAC1 included 457 molecules, and the predicted activation z-score was 2.20 in intestinal-type GC, whereas that in diffuse-type GC was −2.01 (Figure 1b). The causal network of mir-196 included 145 molecules, and the predicted activation z-score was 3.41 in intestinal-type GC, whereas that in diffuse-type GC was −6.73 (Figure 1c). The causal network of ERBB2 included 131 molecules, and the predicted activation z-score was 2.53 in intestinal-type GC, whereas that in diffuse-type GC was −5.68 (Figure 1d).

### 2.2. HDAC1-Interacting Network and Involvement of the Regulation of EMT by the Growth Factors Pathway and the Coronavirus Pathogenesis Pathway

The network pathway analysis of diffuse- and intestinal-type GC identified causal networks of HDAC1 with depth 3, HDAC5, Hdac1/2, HDAC1 with depth 2, HDAC10, and HDAC2 (Table 1). The HDAC1-regulated causal network with depth 3 had 117 regulators in the network and 878 dataset genes downstream of the regulators. The HDAC1-regulated causal network had 115 target-connected regulators, which interact with at least one target molecule in the dataset. Among the causal networks of HDAC molecules, the HDAC1-regulated causal network with depth two was inhibited (activation z-score: −2.011) in diffuse-type GC and activated (activation z-score: 2.199) in intestinal-type GC (Table 1).

HDAC1 was predicted to be inhibited in the causal network of HDAC1 with depth 2 in diffuse-type GC, whereas HDAC1 was predicted to be activated in intestinal-type GC (Figure 2). As shown in Figure 2, the HDAC1-regulated causal network had two layers of regulators, which consisted of epidermal growth factor receptor (EGFR), HDAC2, lymphoid enhancer binding factor 1 (LEF1), E2F transcription factor 3 (E2F3), peroxisome proliferator activated receptor gamma (PPARG), tumor protein p53 (TP53), eukaryotic elongation factor 2 kinase (EEF2K), androgen receptor (AR), signal transducer and activator of transcription (STAT), lysine demethylase 1A (KDM1A), SMAD family member 7 (SMAD7), Smad2/3-Smad4, nuclear factor kappa B (NFkB) (complex), and RB transcriptional corepressor 1 (RB1). Targets of the causal network included 457 molecules, of which 26 molecules (caveolae associate protein 2 (CAVIN2), cyclin E1 (CCNE1), CD34 molecule (CD34), cell division cycle 25A (CDC25A), cyclin dependent kinase 2 (CDK2), chromatin licensing and DNA replication factor 1 (CDT1), centromere protein X (CENPX), DENN domain containing 2B (DENND2B), E2F transcription factor 3 (E2F3), family with sequence similarity 107 member A (FAM107A), GLI family zinc finger 1 (GLI1), HDAC2, minichromosome maintenance 10 replication initiation factor (MCM10), MCM3, MCM7, myocyte enhancer factor 2C (MEF2C), MYB proto-oncogene like 2 (MYBL2), nuclear autoantigenic sperm protein (NASP), paired box 8 (PAX8), DNA polymerase delta 2, accessory subunit (POLD2), DNA primase subunit 2 (PRIM2), RAD54 like (RAD54L), SATB homeobox 1 (SATB1), SRY-box transcription factor 9 (SOX9), syntaxin 1A (STX1A), TPX2 microtubule nucleation factor (TPX2)) were regulated by HDAC1.

HDAC1-interacting network identified the involvement of the regulation of EMT by the growth factors pathway, coronavirus pathogenesis pathway, and vorinostat (Figure 3). HDAC1 was predicted to be activated in diffuse- and intestinal-type GC (Figure 3). Vorinostat is an HDAC class I inhibitor, which regulates cyclin dependent kinase inhibitor 1A (CDKN1A), BCL2 like 1 (BCL2L1), Histone h3, BCL2L11, TP53, Histone h4, CASP8 and FADD like apoptosis regulator (CFLAR), cyclin D1 (CCND1), BCR activator of RhoGEF and GTPase—ABL proto-oncogene 1, non-receptor tyrosine kinase (BCR-ABL1), Raf-1 proto-oncogene, serine/threonine kinase (RAF1), tumor necrosis factor receptor superfamily member 10b (TNFRSF10B), TNF, Akt, BCL2 antagonist/killer 1 (BAK1), and CDKN1B.

The HDAC1-regulated causal network with depth three had RNA–RNA interactions with microRNAs such as mir-10, mir-15, mir-17, mir-19, mir-21, mir-223, mir-25, mir-27, mir-29, and mir-34 (Figure 4). The HDAC-regulated causal network with depth three had 115 target-connected regulators, shown with activation prediction in diffuse-type and intestinal-type GC in Figure 4.

### 2.3. Regulator Effect Network of Diffuse-Type Gastric Cancer

The regulator effect networks of diffuse-type GC included infection of cells (Figure 5). The infection of cells, infection by HIV-1, and extrapulmonary squamous cell carcinoma were the main functions related to the network. Trafficking protein particle complex subunit 2 (TRAPPC1), lysine acetyltransferase 2A (KAT2A), and zinc finger protein 768 (ZNF768) were inactivated, while GATA binding protein 1 (GATA1) was activated in the network (Figure 5). The network had a direct relationship between 10 microRNAs, including let-7, mir-15, mir-17, mir-34, mir-8, mir-497, miR-136-3p (miRNAs w/seed AUCAUCG), miR-3529-3p (miRNAs w/seed ACAACAA), miR-3680-3p (miRNAs w/seed UUUGCAU), and miR-7215-5p (miRNAs w/seed CUCUUUA) (Table 2).

### 2.4. Causal Networks of Lenvatinib in Diffuse- and Intestinal-Type GC

The causal networks of lenvatinib in diffuse-type and intestinal-type GC are shown in Figure 6. The network included KIT proto-oncogene, receptor tyrosine kinase (KIT), STAT3, HRas proto-oncogene, GTPase (HRAS), and integrin subunit beta 1 (ITGB1) (Figure 6). The lenvatinib was predicted as being activated and inactivated in diffuse-type GC and intestinal-type GC, respectively (Figure 6).

## 3. Discussion

Two distinct molecular subtypes, the mesenchymal and epithelial phenotypes, in GC have been identified [15]. The mesenchymal-like type includes diffuse type with poor prognosis [16]. The TP53-active and TP53-inactive types include patients with an intermediate prognosis, which raises an important issue that the subtypes with molecular hallmarks are risk factors of prognosis [16].

Lenvatinib, pyrotinib, HDAC1, mir-196, and ERBB2 have been identified as master regulators of causal networks in diffuse- and intestinal-type GC in the analysis. Since diffuse-type GC has relatively stable genomic features, the development of targeting therapies has been challenging [17]. The combination of focal adhesion kinase inhibitor and mitogen-activated protein kinase (MAPK) kinase inhibitor was effective in inhibiting the tumor growth of human diffuse-type GC xenograft [17]. Lenvatinib, a multi-kinase inhibitor to inhibit vascular endothelial growth factor receptor (VEGFR), fibroblast growth factor receptor (FGFR), platelet-derived growth factor receptor (PDGFR) alpha, KIT, and rearranged during transfection (RET), showed an effect in the progression-free survival and response rate in patients with radioiodine-refractory thyroid cancer [18]. Pyrotinib is an inhibitor of EGFR (ERBB1) and HER2/4 (ERBB2/4), which is approved for the treatment of breast cancer in China [19]. Since both lenvatinib- and pyrotinib-oriented causal networks in diffuse-type GC were activated, it may also be targeted in the treatment of diffuse-type GC. The effects of the combination therapy with sorafenib and lenvatinib were limited in hepatocellular carcinoma. Careful examination in terms of drug resistance and effectiveness is required [20].

Histone modification is involved in drug resistance in lung cancer [21]. Long non-coding RNA HRCEG, which HDAC1 regulates, inhibited cell proliferation and EMT in GC [22]. Silencing of HDAC1 inhibited the proliferation of GC, which suggested the role of HDAC1 as a target for the treatment of GC [23]. HDAC expression determines the sub-type of GC and is involved in tumor microenvironment characteristics and immunotherapy efficacy in GC [24]. Class I HDAC inhibitor induced lipid peroxidation and ferroptosis, which inhibit tumor cell growth [25]. HDAC inhibitors demonstrate an anti-cancer effect via the production of reactive oxygen species, of which dampening renders the resistance to HDAC inhibitors in cancer cells, which requires future investigation to reveal the mechanism of resistance acquisition of cancer [26,27].

The causal network of mir-196, a master regulator of the network, was activated in intestinal-type GC and inactivated in diffuse-type GC. Inhibitor of growth family member 5 (ING5), a class II tumor suppressor, is translationally targeted by miR-196, miR-196a, miR-196b-5p, miR-193a-3p, and miR-27-3p [28]. ING5 promotes the autoacetylation of p53 and histone H3 and H4 to induce the transcription of Bax, growth arrest and DNA damage-inducible 45 (GADD45), p21, and p27 [28]. miRNAs are involved in linking obesity and cancer [29]. The investigation of miRNA–miRNA and miRNA–long non-coding RNA interaction revealed the link between PPARgamma cell signaling regulated by miR-130, miR-4663, miR-375, miR-494-3p, and miR-128-3p and MAPK cell signaling regulated by miR-143, miR-375, miR-196, and miR-128-3p [29]. miR-196 is overexpressed in the intestinal epithelium of Crohn’s disease patients, for which the relationship between cancer and Crohn’s disease is unknown [30]. miR-196 is upregulated in pancreatic cancer cells and activates the AKT signaling pathway, which is involved in the development of type 2 diabetes [31]. The expression level of miR-196b was higher in pancreatic cancer cells than in cancer stroma, and the high expression of miR-196b decreased the overall survival rate, which suggested the role of miR-196b as a prognosis biomarker for pancreatic cancer [32]. The relationship between miRNAs and causal networks may need to be further elucidated to reveal the malignancy.

Regulator effect network analysis in diffuse-type GC revealed the relationship between the network of infection and let-7, mir-15, mir-17, mir-34, mir-8, mir-497, miR-136-3p (miRNAs w/seed AUCAUCG), miR-3529-3p (miRNAs w/seed ACAACAA), miR-3680-3p (miRNAs w/seed UUUGCAU), and miR-7215-5p (miRNAs w/seed CUCUUUA). Let-7 plays a crucial role in the development of virus and cancer-associated virus infection [33]. Since let-7 serves as a regulator of several cellular processes [33], it may be challenging to target let-7 in general to treat diseases; some specific targeting for disease-associated cells would be valuable. Let-7 regulates the self-renewal and tumorigenicity of breast cancer cells [34]. Let-7 was decreased in breast tumor-initiating cells and increased with differentiation [34]. Let-7 may be involved in reducing cancer cell resistance to chemotherapy by silencing the target molecules to inhibit the self-renewal [34]. Let-7 has been identified as a starting point of the RNA revolution and a potential target for cancer and immune therapy [35]. These various roles of let-7 are crucial for considering cancer therapeutics.

The limitation of our approach includes that HDAC1, found as one of the master regulators, is just one member of the class I HDACs in the HDAC classification, with HDAC 2 and 3 (and 8) being other members. Since vorinostat inhibits HDACs 1, 2, and 3, the roles of HDACs 2 and 3 might be important in the metastatic GC. Further advancement of studies in other members of class I HDACs is needed to highlight the role of HDAC inhibitors in drug resistance of GC [36]. Panobinostat, an HDAC inhibitor, suppresses the cell proliferation, metastasis, and cell cycle progression of GC cells [37]. Not only vorinostat but other possible HDAC inhibitors may also be helpful in treating GC.

In conclusion, several causal networks in diffuse- and intestinal-type GC have been identified in the study. EMT characterizes metastatic GC, and drug resistance is closely related to EMT. The master regulators of the causal network included lenvatinib, pyrotinib, HDAC1, mir-196, and ERBB2. Our study highlighted the new therapeutics targeting the master regulators of causal networks for metastatic GC characterized by EMT. In the future, further procedures, i.e., further steps including in vitro, ex vivo, and in vivo experiments and clinical studies, are needed to validate the findings of this study in the application of novel therapeutic targets. Furthermore, the approach based on molecular networks may identify possible side effects critical to new drugs.

## 4. Materials and Methods

### 4.1. Data Analysis of Diffuse- and Intestinal-Type GC

We used RNA sequencing data of diffuse- and intestinal-type GC, which are publicly available in The Cancer Genome Atlas (TCGA) of the cBioPortal for Cancer Genomics database (http://www.cbioportal.org, accessed on 6 August 2024) at the National Cancer Institute (NCI) Genomic Data Commons (GDC) data portal (https://portal.gdc.cancer.gov/, accessed on 6 August 2024) [38,39,40,41,42]. Publicly available data on stomach adenocarcinoma in the TCGA, Stomach Adenocarcinoma (TCGA, PanCancer Atlas) [38,39,40,41], were compared between diffuse-type GC, which is genomically stable (*n* = 50), and intestinal-type GC, which has a feature of chromosomal instability (*n* = 223), in TCGA Research Network publications, as previously described [2,3,39].

### 4.2. Network Pathway Analysis

Data on intestinal- and diffuse-type GC from the TCGA cBioPortal for Cancer Genomics were uploaded and analyzed using Ingenuity Pathway Analysis (IPA) (https://digitalinsights.qiagen.com, accessed on 6 August 2024) (QIAGEN Digital Insights, Aarhus C, Denmark) [43]. The 1099 causal networks in diffuse- and intestinal-type GC were generated by filtering with a cut-off z-score absolute value of 2 (As of July 2023).

### 4.3. Data Visualization

The results of the causal network analyses were visualized with Tableau software (Tableau Desktop 2023.3) (https://www.tableau.com, accessed on 18 June 2024) (Salesforce, Inc., San Francisco, CA, USA, Tableau Global Headquarters, Seattle, WA, USA).

### 4.4. Statistical Analysis

The RNA sequencing data on diffuse- and intestinal-type GC were analyzed using Student’s *t*-test. The z-scores of intestinal- and diffuse-type GC samples were compared, and the difference was considered significant at *p* < 0.00001, following previous reports [1,2,3].

## Figures and Tables

**Figure 1 ijms-25-08821-f001:**
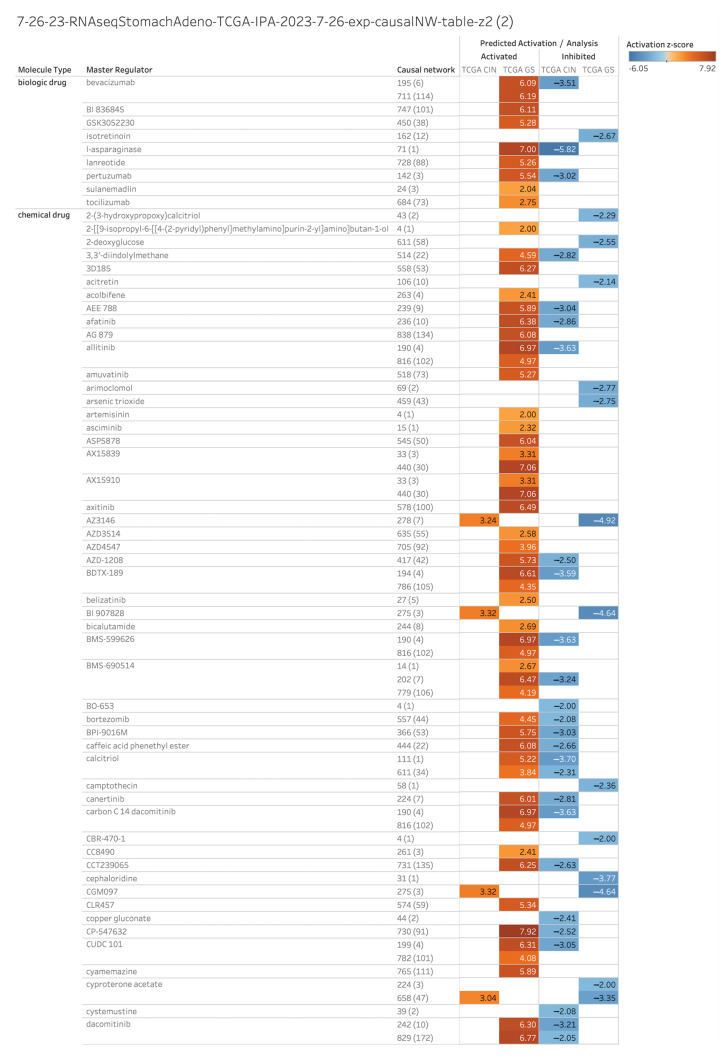
Master regulators of 1099 causal networks in diffuse- and intestinal-type GC included lenvatinib, pyrotinib, HDAC1, mir-196, and ERBB2 (As of July 2023). Lenvatinib, pyrotinib, HDAC1, mir-196, and ERBB2 are indicated with black boxes. (**a**) Biological drug and chemical drug of molecule type. Lenvatinib and pyrotinib are indicated with black boxes; (**b**) growth factor and transcription regulator of molecule type. HDAC1 is indicated with a black box; (**c**) microRNA of molecule type. mir-196 is indicated with a black box; (**d**) kinase of molecule type. ERBB2 is indicated with a black box.

**Figure 2 ijms-25-08821-f002:**
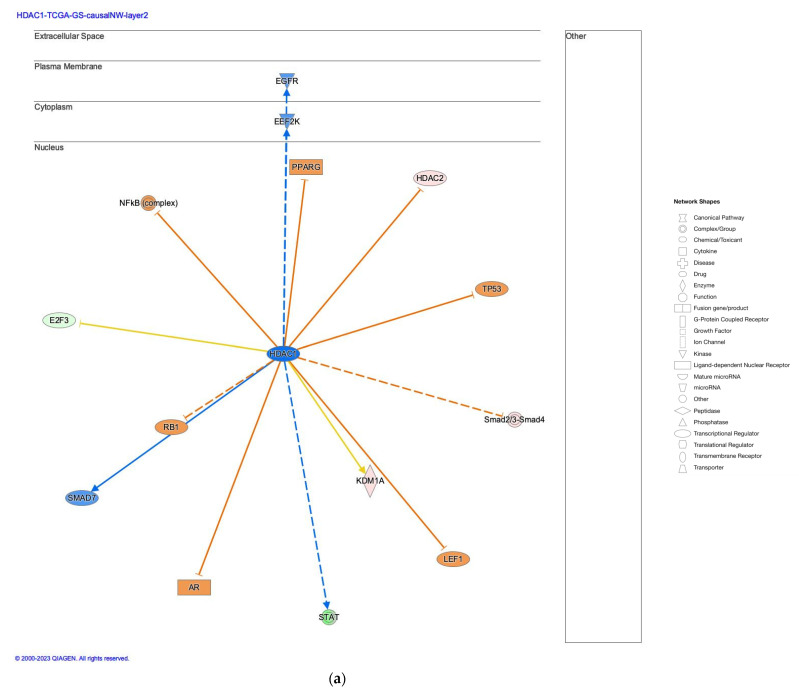
The causal network regulated by HDAC1 with depth two. (**a**) Regulators of the HDAC1-regulated causal network in diffuse-type GC are shown with activation prediction; (**b**) regulators of the HDAC1-regulated causal network in intestinal-type GC are shown with activation prediction. Pale red or green color indicates upregulated or downregulated gene expression, respectively. Orange or blue color indicates predicted activation or inhibition, respectively. The intensity of colors indicates the degree of up- or downregulation. A solid or dashed line indicates direct or indirect interaction, respectively. Triangle, oval, rectangle, and diamond shapes indicate kinase, transcription regulator, ligand-dependent nuclear receptor, and enzyme, respectively.

**Figure 3 ijms-25-08821-f003:**
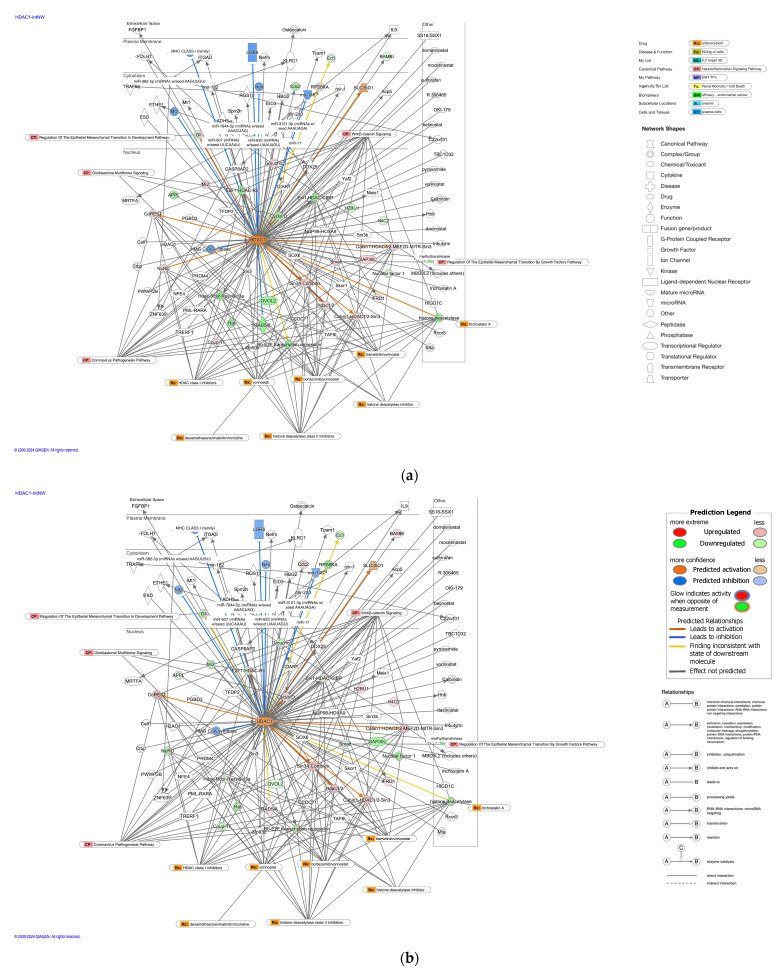
The analysis of the HDAC1-interacting network identified the involvement of the regulation of EMT by the growth factors pathway, coronavirus pathogenesis pathway, and vorinostat. (**a**) The HDAC1-interacting network in diffuse-type GC; (**b**) the HDAC1-interacting network in intestinal-type GC. Pale red or green color indicates upregulated or downregulated gene expression, respectively. Orange or blue color indicates predicted activation or inhibition, respectively. The intensity of colors indicates the degree of up- or downregulation. A solid or dashed line indicates direct or indirect interaction, respectively. (Original images are available in the Appendix A).

**Figure 4 ijms-25-08821-f004:**
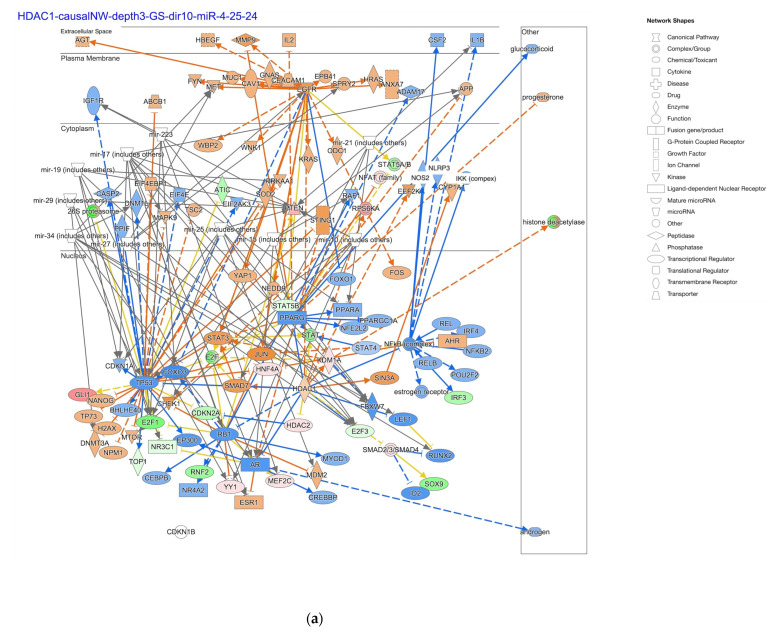
The HDAC1-regulated causal network with depth three had RNA–RNA interactions with microRNAs such as mir-10, mir-15, mir-17, mir-19, mir-21, mir-223, mir-25, mir-27, mir-29, and mir-34. (**a**) The regulators of the HDAC1-regulated causal network with depth three in diffuse-type GC are shown; (**b**) the regulators of the HDAC1-regulated causal network with depth three in intestinal-type GC are shown. Pale red or green color indicates upregulated or downregulated gene expression, respectively. Orange or blue color indicates predicted activation or inhibition, respectively. The intensity of colors indicates the degree of up- or downregulation. A solid or dashed line indicates direct or indirect interaction, respectively. (Original images are available in the Appendix A).

**Figure 5 ijms-25-08821-f005:**
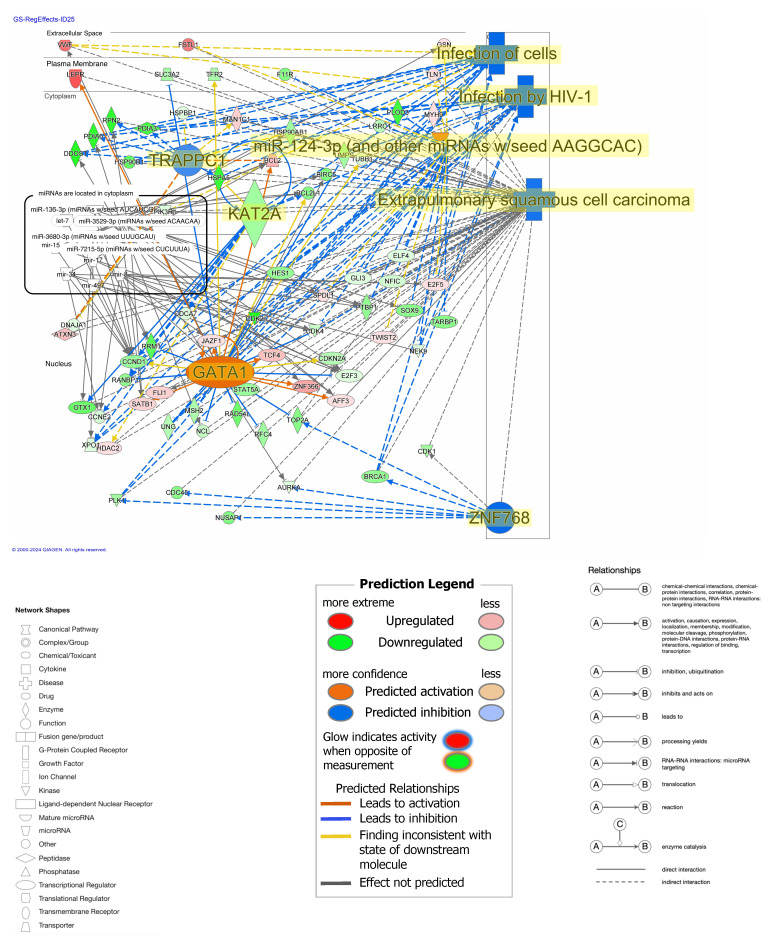
Regulator effect network of diffuse-type GC (Infection of cells, ID25). Pale red or green color indicates upregulated or downregulated gene expression, respectively. Orange or blue color indicates predicted activation or inhibition, respectively. The intensity of colors indicates the degree of up- or downregulation. A solid or dashed line indicates direct or indirect interaction, respectively.

**Figure 6 ijms-25-08821-f006:**
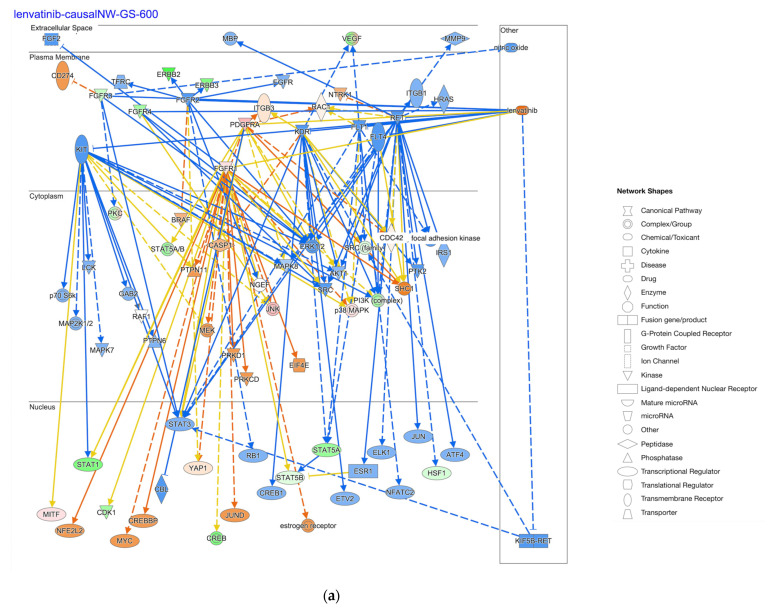
Causal networks of lenvatinib in diffuse- and intestinal-type GC. (**a**) The causal network (depth three) of lenvatinib in diffuse-type GC; (**b**) the causal network (depth three) of lenvatinib in intestinal-type GC. Pale red or green color indicates upregulated or downregulated gene expression, respectively. Orange or blue color indicates predicted activation or inhibition, respectively. The intensity of colors indicates the degree of up- or downregulation. A solid or dashed line indicates direct or indirect interaction, respectively. (Original images are available in the Appendix A).

**Table 1 ijms-25-08821-t001:** Causal networks of HDAC molecules as master regulators in diffuse- and intestinal-type gastric cancer (GC).

Master Regulator	Analysis	Depth	Predicted Activation	Activation z-Score	*p*-Value of Overlap	Network Bias- Corrected *p*-Value	Causal Network	Target-Connected Regulators
HDAC1	TCGA GS	3		0.405	2.87 × 10^−26^	2.00 × 10^−4^	878 (117)	115
	TCGA CIN	3		1.08	2.87 × 10^−26^	2.00 × 10^−4^	878 (117)	115
HDAC5	TCGA GS	3	Inhibited	−4.321	2.01 × 10^−25^	1.00 × 10^−4^	733 (80)	78
	TCGA CIN	3		0.85	2.01 × 10^−25^	1.00 × 10^−4^	733 (80)	78
Hdac1/2	TCGA GS	3	Inhibited	−4.33	2.53 × 10^−25^	1.00 × 10^−3^	820 (95)	95
	TCGA CIN	3		1.606	2.53 × 10^−25^	1.00 × 10^−3^	820 (95)	95
HDAC1	TCGA GS	2	Inhibited	−2.011	7.62 × 10^−19^	1.00 × 10^−4^	457 (15)	15
	TCGA CIN	2	Activated	2.199	7.62 × 10^−19^	1.00 × 10^−4^	457 (15)	15
HDAC10	TCGA GS	3		−0.089	1.43 × 10^−15^	1.70 × 10^−2^	508 (37)	37
	TCGA CIN	3		−1.509	1.43 × 10^−15^	1.70 × 10^−2^	508 (37)	37
HDAC2	TCGA GS	1		1.528	3.55 × 10^−3^	2.24 × 10^−2^	21 (1)	1
	TCGA CIN	1		−0.655	3.55 × 10^−3^	2.24 × 10^−2^	21 (1)	1

TCGA GS (genomically stable): diffuse-type GC, TCGA CIN (chromosome instability): intestinal-type GC.

**Table 2 ijms-25-08821-t002:** MicroRNAs that have a direct relationship with the regulator effect network of diffuse-type gastric cancer.

microRNA	Location
let-7	Cytoplasm
mir-15	Cytoplasm
mir-17	Cytoplasm
mir-34	Cytoplasm
mir-8	Cytoplasm
mir-497	Cytoplasm
miR-136-3p (miRNAs w/seed AUCAUCG)	Cytoplasm
miR-3529-3p (miRNAs w/seed ACAACAA)	Other
miR-3680-3p (miRNAs w/seed UUUGCAU)	Cytoplasm
miR-7215-5p (miRNAs w/seed CUCUUUA)	Cytoplasm

## Data Availability

The original contributions presented in the study are included in the article/Appendix A, further inquiries can be directed to the corresponding author.

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
