# Peer review of "Master Regulators of Causal Networks in Intestinal- and Diffuse-Type Gastric Cancer and the Relation to the RNA Virus Infection Pathway"

_ijms, 2024, doi:10.3390/ijms25168821_

Round 1

Reviewer 1 Report

Comments and Suggestions for Authors

1- Many figures are too small, showing labelings that cannot be read. This applies for example to Figure 1 and Figure 2a,b which are thus not informative. It is appreciated that for example in the case of Figure 3, Figure 4 and Figure 7 it will be barely possible to enlarge it in the main text in a way that details can be read; however, the figures should then at least be given also as Suppl. Material in an enlarged version. In Figure 6, the key information (i.e., the information written in larger letters) is hard to read and should be highlighted.

2- In Figure 5a, essentially no differences in gene expression levels are shown between gastric cancer (organoids) and normal stomach. Please explain the relevance of this figure and where, for example, the Figure shows that “Gene expression of HDAC1 was up-regulated in diffuse-type GC compared to MSCs”, as stated in the text. The text also mentions Figures 5C and 5D, but these figures are not present in the paper. In its current form, the Figure 5 does not seem to support the conclusions of the paper, by rather showing the opposite, i.e., no alterations.

3- In the Abstract, the authors state that the molecular networks revealed in this study may lead to the identification of drug targets for GC. Later in the paper, they specifically mention treatment-resistant cancer. Treatment resistance, however, has not been studied in this paper. The authors’ statement seems to rest only on the fact that metastatic GC is characterized by EMT and drug resistance is closely related to EMT, as stated in the paper. While this may be true, this does not mean that this paper really compares treatment-resistant vs. treatment-sensitive GC. The authors should carefully revise their conclusions and to what extent those are appropriate, and modify the text accordingly.

4- The paper gives the impression that these analyses may more or less directly lead to the identification of novel therapeutic targets. This is probably not so. The authors should clearly describe and discuss the further procedure, i.e. further steps critically required for validation. These may include in vitro, ex vivo, in vivo experiments and clinical studies. Also, the paper fails to mention other aspects important in the identification of novel drugs, e.g., possible side effects. Indeed, vorinostat is not new in the context of GC (albeit one may get this impression from the paper), but one major issue is its side effects.

5- The authors should clearly discuss limitations of their approach, since their findings probably give a not entirely correct impression so far of what’s important and what’s less important. For example, the authors highlight HDAC1, while other HDACs are mentioned much less prominently. HDAC1, however, is just one member of the class I HDACs in the HDAC classification, with HDAC2, 3 (and 8) being other members. It seems that HDAC1 is the main finding n this study not because of its higher relevance but because the other members have been studied less excessively so far. See e.g. doi: 10.3390/cancers14215472. Also note that Vorinostat inhibits HDACs 1, 2, and 3.

6- Likewise, the authors highlight for example Vorinostat, while the probably more efficient inhibitor Panobinostat is not mentioned at all. One reason may be that Panobinostat has been studied less extensively so far. Again, the paper may give a not entirely realistic picture of what’s important and what’s less/not important.

Author Response

Comments 1: Many figures are too small, showing labelings that cannot be read. This applies for example to Figure 1 and Figure 2a,b which are thus not informative. It is appreciated that for example in the case of Figure 3, Figure 4 and Figure 7 it will be barely possible to enlarge it in the main text in a way that details can be read; however, the figures should then at least be given also as Suppl. Material in an enlarged version. In Figure 6, the key information (i.e., the information written in larger letters) is hard to read and should be highlighted. 
Response 1: Thank you for pointing this out. We made Figure 1 and Figure 2a, b larger as much as possible. Figure 3, Figure 4 and Figure 6 (originally Figure 7) are also given in Supplementary Materials as Figure S1-S6. The key information written in larger letters are highlighted in Figure 5 (originally Figure 6) revised.

Comments 2: In Figure 5a, essentially no differences in gene expression levels are shown between gastric cancer (organoids) and normal stomach. Please explain the relevance of this figure and where, for example, the Figure shows that “Gene expression of HDAC1 was up-regulated in diffuse-type GC compared to MSCs”, as stated in the text. The text also mentions Figures 5C and 5D, but these figures are not present in the paper. In its current form, the Figure 5 does not seem to support the conclusions of the paper, by rather showing the opposite, i.e., no alterations. 
Response 2: Agree. Thank you very much for pointing this out. We have, accordingly, revised our manuscript not to include Figure 5 and 2.3. HDAC expression cluster analysis. Figure 6 and Figure 7 have been changed to Figure 5 and Figure 6, respectively. 2.4. Regulator Effect Network of Diffuse-type Gastric Cancer and 2.5. Causal Networks of Lenvatinib in Diffuse- and Intestinal-type GC were changed to 2.3. Regulator Effect Network of Diffuse-type Gastric Cancer and 2.4. Causal Networks of Lenvatinib in Diffuse- and Intestinal-type GC, respectively.

Comments 3: In the Abstract, the authors state that the molecular networks revealed in this study may lead to the identification of drug targets for GC. Later in the paper, they specifically mention treatment-resistant cancer. Treatment resistance, however, has not been studied in this paper. The authors’ statement seems to rest only on the fact that metastatic GC is characterized by EMT and drug resistance is closely related to EMT, as stated in the paper. While this may be true, this does not mean that this paper really compares treatment-resistant vs. treatment-sensitive GC. The authors should carefully revise their conclusions and to what extent those are appropriate, and modify the text accordingly.
Response 3: Agree. Conclusions have been revised to emphasize that metastatic GC is characterized by EMT and drug resistance is closely related to EMT, and this study may lead to the identification of drug targets for GC in 3. Discussion, lines 264-267 of the revised manuscript.
“Lines 264-267: EMT characterizes metastatic GC, and drug resistance is closely related to EMT. The master regulators of the causal network included lenvatinib, pyrotinib, HDAC1, mir-196, and ERBB2. Our study highlighted the new therapeutics targeting the master regulators of causal networks for metastatic GC characterized by EMT.”

Comments 4: The paper gives the impression that these analyses may more or less directly lead to the identification of novel therapeutic targets. This is probably not so. The authors should clearly describe and discuss the further procedure, i.e. further steps critically required for validation. These may include in vitro, ex vivo, in vivo experiments and clinical studies. Also, the paper fails to mention other aspects important in the identification of novel drugs, e.g., possible side effects. Indeed, vorinostat is not new in the context of GC (albeit one may get this impression from the paper), but one major issue is its side effects.
Response 4: Agree. We added the description of the further procedure (further steps critically required for validation in 3. Discussion, in lines 269-271 of the revised manuscript. The important aspects in the identification of novel drugs, possible side effects, were added in 3. Discussion, in lines 271-272.
“Lines 269-272: In the future, further procedures, i.e., further steps including in vitro, ex vivo, in vivo experiments, and clinical studies, are needed to validate the findings of this study in the application of novel therapeutic targets. Furthermore, the approach based on molecular networks may identify possible side effects critical to new drugs..”

Comments 5: The authors should clearly discuss limitations of their approach, since their findings probably give a not entirely correct impression so far of what’s important and what’s less important. For example, the authors highlight HDAC1, while other HDACs are mentioned much less prominently. HDAC1, however, is just one member of the class I HDACs in the HDAC classification, with HDAC2, 3 (and 8) being other members. It seems that HDAC1 is the main finding in this study not because of its higher relevance but because the other members have been studied less excessively so far. See e.g. doi: 10.3390/cancers14215472. Also note that Vorinostat inhibits HDACs 1, 2, and 3.
Response 5: We added the limitation of our approach in 3. Discussion, in lines 255-260.
“Lines 255-260: The limitation of our approach includes that HDAC1, found as one of the master regulators, is just one member of the class I HDACs in the HDAC classification, with HDAC2, 3 (and 8) being other members. Since vorinostat inhibits HDACs 1, 2, and 3, the roles of HDACs 2 and 3 might be important in the metastatic GC. Further advancement of studies in other members of class I HDACs is needed to highlight the role of HDAC inhibitors in drug resistance of GC [36].”
Reference suggested was added as reference 36:
Badie, A.; Gaiddon, C.; Mellitzer, G. Histone Deacetylase Functions in Gastric Cancer: Therapeutic Target? Cancers (Basel) 2022, 14, doi:10.3390/cancers14215472.

Comments 6: Likewise, the authors highlight for example Vorinostat, while the probably more efficient inhibitor Panobinostat is not mentioned at all. One reason may be that Panobinostat has been studied less extensively so far. Again, the paper may give a not entirely realistic picture of what’s important and what’s less/not important.
Response 6: Description of Panobinostat was added in 3. Discussion, in lines 260-262.
“Lines 260-262: Panobinostat, an HDAC inhibitor, suppresses the cell proliferation, metastasis, and cell cycle progression of GC cells [37]. Not only vorinostat but other possible HDAC inhibitors may also be helpful in treating GC.”
Reference 37 was added:
Lee, N.R.; Kim, D.Y.; Jin, H.; Meng, R.; Chae, O.H.; Kim, S.H.; Park, B.H.; Kim, S.M. Inactivation of the Akt/FOXM1 Signaling Pathway by Panobinostat Suppresses the Proliferation and Metastasis of Gastric Cancer Cells. Int J Mol Sci 2021, 22, doi:10.3390/ijms22115955.

Reviewer 2 Report

Comments and Suggestions for Authors

This article provides an informative and sophisticated analysis, identifying master regulators of signaling networks in gastric cancer. Importantly, it offers plausible drug targets to enhance the therapeutic effects of GC treatments. The suggestion to repurpose lenvatinib and pyrotinib is a particularly useful finding. I also appreciate the specific recommendations for improving new HDAC inhibitors compared to existing ones, emphasizing the need for greater specificity and potential combination with other therapies. Additionally, the discussion on the necessity for further investigation into the mechanisms behind resistance to HDAC inhibitors in cancer therapy, and the focus on specific mechanisms, is valuable to the research and medical community. I recommend the publication of this paper. 

Comments on the Quality of English Language

This manuscript is not particularly easy to understand, not due to grammatical errors, but because of stylistic reasons and the complexity of the content. If possible, I recommend using more active voice and breaking down the sentences into shorter ones.

Author Response

Comment 1: This article provides an informative and sophisticated analysis, identifying master regulators of signaling networks in gastric cancer. Importantly, it offers plausible drug targets to enhance the therapeutic effects of GC treatments. The suggestion to repurpose lenvatinib and pyrotinib is a particularly useful finding. I also appreciate the specific recommendations for improving new HDAC inhibitors compared to existing ones, emphasizing the need for greater specificity and potential combination with other therapies. Additionally, the discussion on the necessity for further investigation into the mechanisms behind resistance to HDAC inhibitors in cancer therapy, and the focus on specific mechanisms, is valuable to the research and medical community. I recommend the publication of this paper. 
Response 1: Thank you very much for your nice appreciation of our paper. Your comments are much appreciated.

Comment 2: This manuscript is not particularly easy to understand, not due to grammatical errors, but because of stylistic reasons and the complexity of the content. If possible, I recommend using more active voice and breaking down the sentences into shorter ones.
Response 2: Thank you very much for the suggestions. The manuscript was revised to use more active voice and shorten the sentences.

Round 2

Reviewer 1 Report

Comments and Suggestions for Authors

The authors have sufficiently addressed my previous issues and concerns.